# Geographic Variation and Risk Factor Association of Early Versus Late Onset Colorectal Cancer

**DOI:** 10.3390/cancers15041006

**Published:** 2023-02-04

**Authors:** Weichuan Dong, Uriel Kim, Johnie Rose, Richard S. Hoehn, Matthew Kucmanic, Kirsten Eom, Shu Li, Nathan A. Berger, Siran M. Koroukian

**Affiliations:** 1Population Cancer Analytics Shared Resource and Department of Population and Quantitative Health Sciences, Case Western Reserve University School of Medicine, Cleveland, OH 44106, USA; 2Kellogg School of Management, Northwestern University, Evanston, IL 60208, USA; 3Center for Community Health Integration, Case Western Reserve University School of Medicine, Cleveland, OH 44106, USA; 4Case Comprehensive Cancer Center, Case Western Reserve University School of Medicine, Cleveland, OH 44106, USA; 5Department of Surgery, Division of Surgical Oncology, University Hospitals Cleveland Medical Center, Cleveland, OH 44106, USA; 6Department of Geographical and Sustainability Sciences, University of Iowa, Iowa City, IA 52242, USA; 7MetroHealth Cancer Center, Cleveland, OH 44109, USA; 8School of Digital Sciences, Kent State University, Kent, OH 44240, USA; 9Center for Science, Health and Society, Case Western Reserve University School of Medicine, Cleveland, OH 44106, USA

**Keywords:** colorectal cancer, risk factor, early-onset, geographic information system, machine learning, random forest, regionalization

## Abstract

**Simple Summary:**

While the incidence of late-onset colorectal cancer (LOCRC) has steadily decreased, the incidence of early-onset colorectal cancer (EOCRC) has continued to increase in the US. This study aims to uncover geographic disparities in EOCRC and understand how risk factors between EOCRC and LOCRC differ. The geographic analysis revealed regions with relatively low LOCRC rates and high EOCRC rates, identifying regions with a disproportionate burden of EOCRC. We then evaluated and compared community-level risk factors associated with incidence rates of EOCRC and LOCRC using the random forest machine learning method. The analysis identified a set of risk factors most predictive of EOCRC and LOCRC, such as diabetes prevalence and physical inactivity, but the importance of these risk factors varied between EOCRC and LORC. Collectively, these findings can help facilitate future studies that further uncover actionable interventions to reduce EOCRC and guide where targeted interventions to reduce EOCRC burden should be deployed.

**Abstract:**

The proportion of patients diagnosed with colorectal cancer (CRC) at age < 50 (early-onset CRC, or EOCRC) has steadily increased over the past three decades relative to the proportion of patients diagnosed at age ≥ 50 (late-onset CRC, or LOCRC), despite the reduction in CRC incidence overall. An important gap in the literature is whether EOCRC shares the same community-level risk factors as LOCRC. Thus, we sought to (1) identify disparities in the incidence rates of EOCRC and LOCRC using geospatial analysis and (2) compare the importance of community-level risk factors (racial/ethnic, health status, behavioral, clinical care, physical environmental, and socioeconomic status risk factors) in the prediction of EOCRC and LOCRC incidence rates using a random forest machine learning approach. The incidence data came from the Surveillance, Epidemiology, and End Results program (years 2000–2019). The geospatial analysis revealed large geographic variations in EOCRC and LOCRC incidence rates. For example, some regions had relatively low LOCRC and high EOCRC rates (e.g., Georgia and eastern Texas) while others had relatively high LOCRC and low EOCRC rates (e.g., Iowa and New Jersey). The random forest analysis revealed that the importance of community-level risk factors most predictive of EOCRC versus LOCRC incidence rates differed meaningfully. For example, diabetes prevalence was the most important risk factor in predicting EOCRC incidence rate, but it was a less important risk factor of LOCRC incidence rate; physical inactivity was the most important risk factor in predicting LOCRC incidence rate, but it was the fourth most important predictor for EOCRC incidence rate. Thus, our community-level analysis demonstrates the geographic variation in EOCRC burden and the distinctive set of risk factors most predictive of EOCRC.

## 1. Introduction

Colorectal cancer (CRC) incidence in the US has declined steadily since the mid-1990s, owing to substantial drops in new cases among individuals aged 50 years and older (late-onset CRC, or LOCRC) [1,2]. This trend is likely attributable to improved screening rates in this age group. However, diagnosis of CRC in individuals younger than 50 years old (early-onset CRC, or EOCRC) has nearly doubled over the same period and continues to increase by approximately 2% annually [1]. Thus, uncovering risk factors for EOCRC, especially those that may be unique to early-onset disease versus late-onset disease, can inform efforts that facilitate the prevention and detection of this growing cancer subtype.

Researchers have identified multiple shared risk factors for EOCRC and LOCRC, including sedentary behaviors, Western-style diets, smoking, and diabetes [3,4,5]. Studies suggest that EOCRC may have distinct clinical and molecular phenotypes compared with LOCRC. This may be attributable to differences in demographic and genetic factors, or potentially modifiable behavioral and environmental factors [6]. Studies also have linked CRC risk with social determinants of health [7], which are often measured by area-level characteristics of individuals’ residences as a surrogate. However, the extent to which these factors are associated with EOCRC, and whether they differ from those associated with LOCRC, remains unclear.

Since incidence rates in many sparsely populated counties are often unreliable due to the small number of cases in those areas, it is challenging to conduct population and geographic studies on EOCRC. To our knowledge, no study has conducted a comprehensive area-level analysis comparing demographic, health status, behavioral, clinical care, environmental, and socioeconomic risk factors between EOCRC and LOCRC. This cross-sectional study applied a geographic method (Max-p-regions) to address the data scarcity problem at the county level, followed by comparing the differences in the geographic distributions of the incidence rates of EOCRC and LOCRC. Finally, the study evaluated and compared risk factor importance in the prediction of EOCRC and LOCRC incidence rates.

## 2. Materials and Methods

### 2.1. Study Population and Data Sources

The study included individuals diagnosed with CRC before 50 years of age (EOCRC) and at 50 years or older (LOCRC) between 2000 and 2019 from the Surveillance, Epidemiology, and End Results (SEER) database [8]. Data in the SEER program cover 48% of the U.S. population with 97% completeness within the SEER regions [9]. Geographic regions included in this study cover the U.S. states of California, Connecticut, Georgia, Hawaii, Idaho, Illinois, Iowa, Kentucky, Louisiana, Massachusetts, New Jersey, New Mexico, New York, Texas, Utah, and Washington (Seattle–Puget Sound), with a total of 1085 counties. All data were obtained at the county level and were aggregated into a larger geographic unit for analysis as necessary using the max-p-regions method described in further detail below.

### 2.2. Variables of Interest

The outcomes of interest were the age-adjusted EOCRC incidence rate and age-adjusted LOCRC incidence rate per 100,000 persons. The SEER*STAT software was used to query and extract cancer incidence data and patient characteristics. Predictor variables for the random forest models were obtained from the County Health Rankings and Roadmaps (CHR) [10] and the Health Resources and Services Administration—Area Health Resources Files (AHRF) [11] to include the relevant domains of population race/ethnicity, health status, health behaviors, clinical care, physical environment, and socioeconomic status (Table 1).

### 2.3. Analytic Approach

#### 2.3.1. Composite Counties

To protect patient confidentiality and mitigate unstable cancer incidence rates, counties with fewer than 16 EOCRC or LOCRC cases are typically suppressed. However, since over one-third of counties had fewer than 16 EOCRC or LOCRC cases over the study period, excluding these counties would likely introduce bias by excluding patients from less populated areas. To address this issue, we used a regionalization method called Max-p-Regions [12] to aggregate adjacent, socio-demographically similar counties into the smallest geographic area based on the threshold constraint of at least 16 EOCRC or LOCRC cases and called the newly created area composite county (CC). The county-level area deprivation index (ADI), an index of social deprivation calculated from 17 census variables [13], was used to determine which counties should be grouped together considering their socio-demographical similarity. Next, the age-adjusted incidence rates of EOCRC and LOCRC were calculated at the CC-level. To account for differences in population size of CCs, county-level predictor variables were weighted at the CC-level based on the relative population of the grouped counties.

#### 2.3.2. Random Forest and Variable Importance

Random forest analysis was used to evaluate variable importance (VI) of risk factors in predicting EOCRC incidence rate and LOCRC incidence rate, respectively. Random forest is a tree-based, nonlinear, nonparametric machine learning method that creates and aggregates an ensemble of trees for prediction using random variable selection and bootstrap sampling [14]. In each regression tree of the random forest, every predictor is “competing” with all other predictors during the selection of a split node. The results of the random forest analysis are highly stable and replicable because they are based on the average output of all its regression trees [14]. Compared with traditional linear regression models, random forest can evaluate a large number of predictors without concerns about correlation among them, as demonstrated in a previous study [15].

VI is a measure of node impurity, which reflects the extent to which stratification by a given variable minimizes the variance of responses within resultant subgroups in the regression trees of the random forest. Generally, the frequency of a predictor serving as the splits of trees in the random forest determines its VI. The VI of each predictor was then ordered from highest to lowest, and was also classified into high, median high, median low, and low using the Jenks natural breaks classification method, a data classification method that reduces the variance within classes and maximizes the variance between classes [16].

We created 20,000 trees with all variables from Table 1 included as predictors in the random forest analyses. The number of variables randomly sampled as candidates at each tree split was set to 5. R statistical software (version 4.2.1) and the package “randomForest” were used for the random forest analyses.

Because the random forest is a nonlinear tree-based method and cannot determine the direction of association between the outcome and predictors, we used the correlation coefficient to represent the direction of association as a surrogate (i.e., positive/negative, or +/−) in the VI plots.

## 3. Results

### 3.1. Patient Characteristics of EOCRC and LOCRC

Table 2 shows the patient characteristics of EOCRC and LOCRC in the study area. Our study included 136,065 and 1,141,775 individuals who were newly diagnosed with invasive EOCRC and LOCRC, respectively, in the 1085 SEER counties. The overall EOCRC incidence rate and LOCRC incidence rate were 6.9 and 134.7 per 100,000 persons, respectively. Of those, 47.7% of EOCRC and 48.4% of LOCRC patients were women. Non-Hispanic White was the most common race/ethnicity group for both EOCRC and LOCRC patients. Compared to LOCRC, there were more EOCRC patients who were Non-Hispanic Black (13.9% vs. 11.2%), Non-Hispanic Asian or Pacific Islander (7.5% vs. 5.5%), and Hispanic (18.5% vs. 10.4%), while there were fewer EOCRC patients who were Non-Hispanic White (58.7% vs. 72.2%). The number of EOCRC cases increased substantially every 5 years, with a total increase of 31.6% from 2000–2004 to 2015–2019, while the number of LOCRC cases decreased by 10.1% over the same period. Regarding tumor site, there were more EOCRC patients who had tumors in the rectum or rectosigmoid junction compared to LOCRC patients (38.4% vs. 27.9%).

### 3.2. Geographic Distribution of EOCRC and LOCRC

The max-p-regions method created 709 CCs from the 1085 counties. The medians (IQR) of CC-level EOCRC and LOCRC incidence rates were 7.4 (6.6–8.5) and 142.0 (129.4–156.9) per 100,000 persons, respectively. Figure 1 shows the geographic distributions of the CC-level incidence rates of EOCRC and LOCRC by quartile in the SEER regions. Generally, states in the West (California, Washington (Seattle–Puget Sound), Idaho, Utah, and New Mexico) had lower EOCRC and LOCRC incidence rates than those in the Midwest (Iowa and Illinois) or the South (Kentucky, Louisiana, and Georgia). Kentucky, Louisiana, and Southern Illinois had the highest incidence rates of both EOCRC and LOCRC. Notably, Georgia and eastern Texas had relatively higher EOCRC incidence rates and lower LOCRC incidence rates compared to other areas. In contrast, Iowa, New Jersey, northern Illinois, and upstate New York had relatively lower EOCRC incidence rates and higher LOCRC incidence rates compared to other areas.

### 3.3. Importance of Risk Factors in Predicting EOCRC and LOCRC

Figure 2 shows the VI of risk factors from the random forest analyses in predicting EOCRC incidence rate and LOCRC incidence rate. The most important variable is set to 100%, and the VI of the rest of the variables is scaled relative to the most important variable. The position of the horizontal lines (+/−) indicates the direction of association determined by a linear correlation coefficient between the predictor and the outcome. We observed that the direction of association was consistent between the EOCRC and the LOCRC models for all risk factors. Figure 3 presents the differences in the rankings of the variables with the ten highest VI between the EOCRC and the LOCRC models.

For race/ethnicity, the Hispanic population predictor had a high VI (ranked second) in the EOCRC model and medium-high VI (ranked second) in the LOCRC model, with a negative association with both outcomes, meaning that both of these diseases were less likely to be in the Hispanic population. The Non-Hispanic Black predictor was positively associated with both EOCRC and LOCR, but their VI was low for both outcomes (20th and 10th, respectively). The Non-Hispanic White predictor was also positively associated with both EOCRC and LOCRC, and their VI was medium-low for both outcomes (11th for EOCRC and sixth for LOCRC).

For health status risk factors, diabetes prevalence had the highest VI (first) for EOCRC, and it had a medium-low VI (seventh) for LOCRC. Low birthweight was relatively more important in the EOCRC compared to that in the LOCRC model (sixth vs. 16th).

For health behavioral risk factors, both physical inactivity and smoking were positively associated with EOCRC and LOCRC, with physical inactivity ranked fourth for EOCRC and first for LOCRC, and smoking ranked third for both EOCRC and LOCRC. Adult obesity was ranked fifth (medium-low VI) in both models. Excessive drinking was negatively associated with EOCRC and LOCRC and had a medium-low VI (12th) for EOCRC and a low VI for LOCRC (eighth). Other health behavioral risk factors, including access to recreational facilities, food insecurity, insufficient sleep, and teen birth, had a low VI for both EOCRC and LOCRC outcomes.

For clinical care risk factors, the preventable hospital stays predictor was positively associated with both outcomes with a medium-low VI, but its rank of VI increased from 10th for EOCRC to fourth for LOCRC. The predictor of primary care physicians was negatively associated with both outcomes with its VI ranked 10th for EOCRC and 27th for LOCRC. Community health centers and health care cost had a low VI for both EOCRC and LOCRC, despite their positive associations with both outcomes.

Regarding physical environment of the composite counties, air pollution, driving alone to work (indicators of the transit system and physical inactivity), and rural population all had a positive association with both EOCRC and LOCRC outcomes. However, air pollution and driving alone to work had a low VI for both outcomes, and rural population had a medium-low VI (seventh) for EOCRC and a low VI (11th) for LOCRC.

## 4. Discussion

Using geographic and random forest methods, this cross-sectional study explored the geographic distribution and risk factor association of EOCRC incidence rate and LOCRC incidence rate across the US at the community level. We found substantial geographic disparities in the risk of EOCRC and LOCRC. Certain regions also had a relatively low LOCRC risk and high EOCRC risk (e.g., Georgia and eastern Texas) while other areas presented relatively high LOCRC risk and low EOCRC risk (e.g., Iowa and New Jersey). The random forest analyses revealed that certain risk factors, especially diabetes prevalence and physical inactivity, had different degrees of importance in predicting the incidence rate of EOCRC and the incidence rate of LOCRC.

Diabetes is a known risk factor for CRC [17]. For example, studies from the Netherlands and Sweden found that diabetes was associated with an increased risk of CRC for patients at a younger age compared to their older counterparts [18,19]. Reinforcing these previous studies, our study found that diabetes prevalence was the top important variable in predicting the incidence rate of EOCRC, but not a high-importance variable in predicting the incidence rate of LOCRC. An interesting geographic region of investigation for diabetes and its relationship with EOCRC is the “Diabetes Belt”, defined by researchers as a region comprising 644 counties in the American South [20] (and includes the states of Georgia, Kentucky, and Louisiana in our study area). Previous research has also noted that people in the Diabetes Belt were more likely to be non-Hispanic Black, have a sedentary lifestyle, be obese, and have a smaller fitness/recreation facility density than in the rest of the US [20,21]. In our geographic analysis, we found that in Georgia, a state in the Diabetes Belt and whose diabetes prevalence was among the highest, most composite counties were in the top quartile of EOCRC incidence rates but were in lower quartiles of LOCRC incident rates (Figure 1), suggesting that diabetes has a stronger effect on EOCRC versus LOCRC. In recent years, the prevalence of type 2 diabetes in adolescents and young adults has dramatically increased [22,23], and increases have been disproportionally observed among African Americans [23,24]. Thus, future studies should continue to investigate the role of place, race, ethnicity, diabetes, and EOCRC.

Our models identified physical inactivity as having the highest VI for LOCRC and a medium-high VI for EOCRC, consistent with numerous studies suggesting that physical inactivity or sedentary lifestyle is a substantial risk factor for CRC [3,5,25]. Physical inactivity is also associated with other chronic conditions, such as obesity and diabetes [26,27], which may also increase the risk of CRC. In recent years, sedentary-based work practices and passive behaviors such as smartphone use have increased among young adults, especially in the COVID-19 era [28,29]. A growing body of literature has also linked sedentary lifestyle to an increased risk of EOCRC [30,31,32]. Our study provides further evidence suggesting that increasing physical activities may be an effective and actionable risk reduction strategy for CRC for people of all ages.

As a risk factor strongly associated with diabetes and physical inactivity, adult obesity had a relatively low VI in predicting either EOCRC incidence rate or LOCRC incidence rate. Despite CRC being one of the obesity-associated cancers [33,34], associations between obesity and EOCRC have not been consistently shown in earlier studies [3,4,35].

Regarding other health behavioral risk factors, smoking was the third important predictor in both the EOCRC and LOCRC models in our study, which supports earlier findings on smoking being a risk factor for CRC [3,4,5]. Interestingly, excessive drinking was negatively associated with EOCRC and LOCRC in our study, though it should be noted that the risk factor had a relatively low VI. Alcohol consumption, especially heavy drinking, has been associated with increased CRC risk in previous studies [36,37,38]. This inconsistency may be because excessive drinking is confounded by factors such as the rurality of residence and diet. In another study, moderate alcohol consumption was associated with a reduced CRC risk in populations with greater adherence to a Mediterranean diet [39]. The lack of consistency warrants further investigation into the effect of alcohol consumption on the risk of EOCRC and LOCRC.

There were substantial differences in the risk of EOCRC and LOCRC among racial/ethnic groups. Recent epidemiological data show that the incidence of CRC for Hispanics was 25% and 12% lower than that for Non-Hispanic Blacks and Non-Hispanic Whites in the US, respectively [1]. However, this difference narrowed for patients older than 50, as suggested by a population-based study of patients from California [40]. Despite the negative association between Hispanic population and CRC risk from our study, our data in Table 2 show that the proportion of Hispanic patients with EOCRC was much higher than those with LOCRC (18.5% vs. 10.4%). This might be attributable to changes in lifestyle between the US-born Hispanic population versus first-generation Hispanic immigrants; not only are US-born Hispanics younger on average compared to first-generation Hispanic immigrants [41], but they also report lower intake of fruits and vegetables [42], higher rates of smoking [43], higher rates of physical inactivity [44], and higher rates of obesity [45]. Previous studies have also shown that Hispanics born in the US have a higher risk of colorectal cancer death compared to their foreign-born counterparts [46] and even Non-Hispanic Whites [47]. Thus, as the proportion of US-born Hispanics increases and the proportion of foreign-born Hispanics decreases [48], the association between Hispanic ethnicity and CRC burden may evolve [49].

Finally, compared to race/ethnicity, health status, and health behaviors, risk factors related to clinical care, physical environment, and socioeconomic status did not show high VI in predicting either EOCRC or LOCRC incidence rate. However, these risk factors might still influence the risk of EOCRC and LOCRC through their impact on other risk factors such as smoking, physical inactivity, obesity, and diabetes, as suggested by their directions of association in Figure 2. Future studies should investigate the causal pathways between these risk factors and CRC.

Our findings should be interpreted in light of the following limitations. First, we were not able to consider temporal variations in either the outcomes or the predictors during the 20-year study period. Second, this community-level study did not consider individual-level characteristics in the analyses. However, many of the associations identified from our study are consistent with previous individual-level studies on risk factors of EOCRC and LOCRC. Finally, we could not consider spatially varying associations [50] between the risk factors and EOCRC and LOCRC due to the relatively small number of geographic units (composite counties) and non-contiguous regions in the study area.

We note that this random forest approach is data-driven and results are stable and reproducible based on the data inputs. Future studies using new additional data may impact the results of EOCRC and LOCRC. Also, the focus here is not to discover causal pathways for EOCRC or LOCRC, noting that causality cannot be established from this study, given its cross-sectional nature. In addition, the results obtained from data-driven approaches should be interpreted with caution, and the plausibility of the emerging associations should be determined in light of the relevant body of knowledge. Nevertheless, evaluating the association between community-level risk factors and outcomes is an essential first step toward establishing causality.

## 5. Conclusions

This community-level study evaluated and compared associations between various risk factors and EOCRC and LOCRC using novel geographic and machine learning approaches. The geographic analysis revealed regions with an elevated risk of EOCRC but a lower risk of LOCRC, identifying regions with a disproportionate burden of EOCRC. The machine learning analysis identified risk factor profiles specific to EOCRC and LOCRC. Collectively, these findings can help facilitate future studies that further uncover actionable interventions to reduce EOCRC and guide where targeted interventions to reduce EOCRC burden should be deployed.

## Figures and Tables

**Figure 1 cancers-15-01006-f001:**
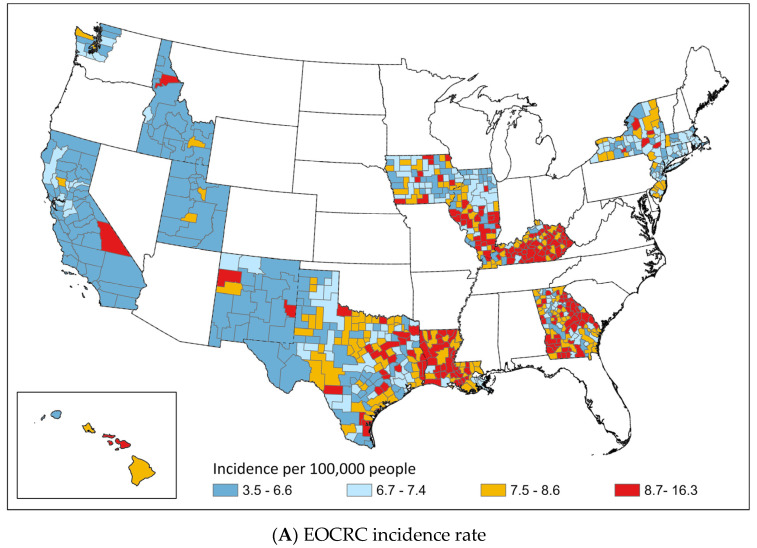
Geographic distribution of (**A**) EOCRC incidence rate, and (**B**) LOCRC incidence rate. Notes: Incidence rates are age-adjusted and classified by quartile. Data for states in white are not available in SEER.

**Figure 2 cancers-15-01006-f002:**
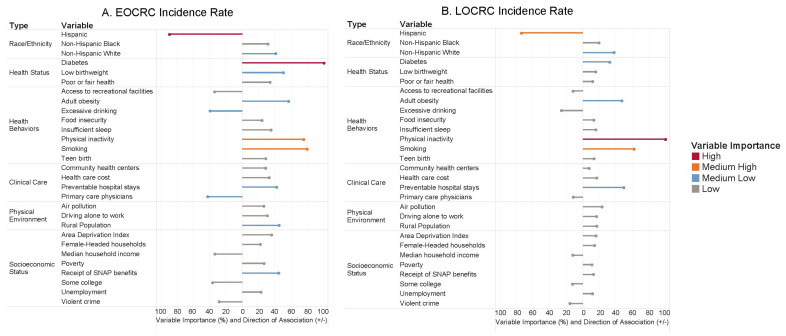
Variable importance and direction of association of risk factors for (**A**) EOCRC incidence rate, and (**B**) LOCRC incidence rate. Notes: (1) The most important variable is set to 100%. The importance of the rest of the variables is scaled relative to the most important variable. (2) Direction of association was determined by a linear correlation coefficient. (3) Categories of variable importance (i.e., high to low) was classified by the Jenks natural breaks method.

**Figure 3 cancers-15-01006-f003:**
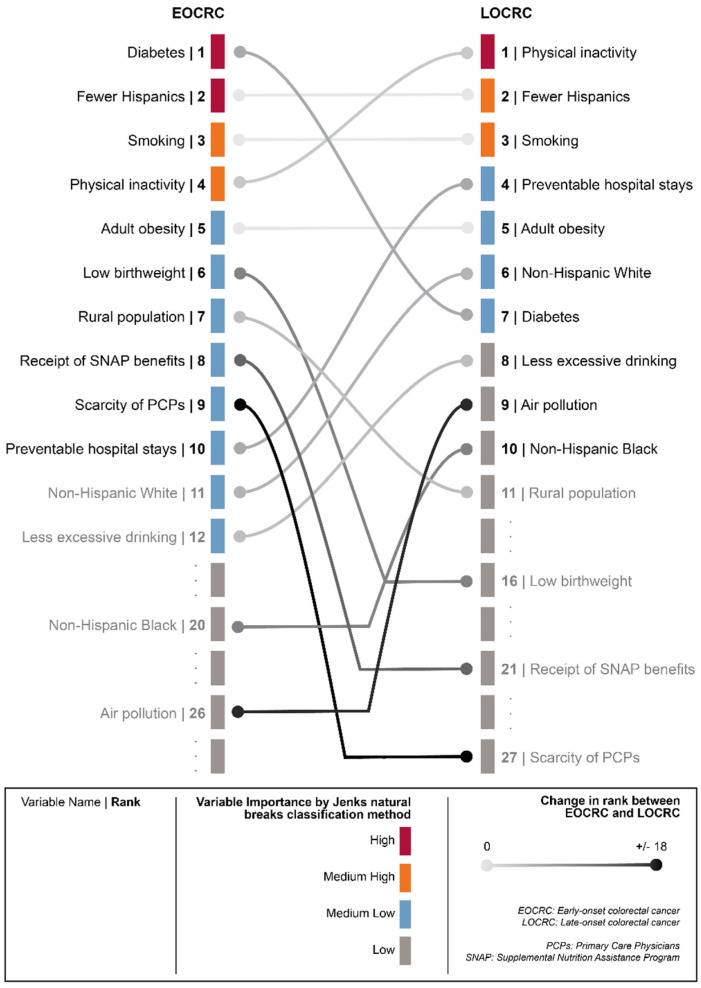
Comparisons of the top ten risk factors based on variable importance between EOCRC and LOCRC models.

**Table 1 cancers-15-01006-t001:** County-level explanatory variables used in random forest analyses.

Variable	Description	Year	Original Source
**Ethnicity/Race**			
Non-Hispanic Black	Percentage of Non-Hispanic African American population	2011	Census—PE
Non-Hispanic White	Percentage of Non-Hispanic White population		
Hispanic	Percentage of population identifying as Hispanics		
**Health Status**			
Diabetes	Percentage of adults (age 20+) with diagnosed diabetes (age-adjusted)	2009	CDC—DIA
Low birthweight	Percentage of live births with low birthweight (<2500 g)	2004–2010	CDC—NCHS
Poor or fair health	Percentage of adults reporting fair or poor health (age-adjusted)	2014	CDC—BRFSS
**Health Behaviors**			
Adult obesity	Percentage of the people (age 18+) that report a body mass index ≥ 30 (age-adjusted)	2009	CDC—DIA
Physical inactivity	Percentage of adults (age 18+) reporting no leisure-time physical activity (age-adjusted)	2009	
Excessive drinking	Percentage of adults reporting binge or heavy drinking (age-adjusted)	2014	CDC—BRFSS
Insufficient sleep	Percentage of adults who report fewer than 7 h of sleep on average (age-adjusted)	2014	
Smoking	Percentage of adults who are current smokers (age-adjusted)	2014	
Access to recreational facilities	Number of recreational facilities (engaged in fitness and recreational sports) per 100,000 persons	2010	USDA FEA and Census CBP
Food insecurity	Percentage of population who lack adequate access to food	2013	MMP
Teen birth	Number of births per 1000 females ages 15–19	2004–2010	CDC—NCHS
**Clinical Care**			
Uninsured adults	Percentage of adults under age 65 without health insurance	2010	Census—SAHIE
Health care costs	Per capita spending of Medicare enrollees	2009	DAHC
Preventable hospital stays	Rate of hospital stays for ambulatory-care sensitive conditions per 100,000 Medicare enrollees	2010	
Primary care physicians	Primary care physicians in patient care per 100,000 persons	2010	HRSA
Community health centers	Community health centers per 100,000 persons	2013	
**Physical Environment**			
Air pollution	Total days when the fine particulate matter (PM2.5) exceeded the 24 h primary standard established by the US EPA.	2008	CDC WONDER
Driving alone to work	Percentage of the workforce that drives alone to work (indicators of the transit system and physical inactivity)	2007–2011	Census—ACS
Rural population	Percentage of people living in rural areas	2010	Census—PE
**Socioeconomic Status**			
Area deprivation index	A composite measure of 17 census variables designed to describe social deprivation	2008–2012	Census—ACS
Median household income	The income where half of households in a county earn more and half of households earn less	2011	Census—SAIPE
Poverty	Percentage of people in poverty	2010	
Receipt of SNAP benefits	Percentage of people who were food stamp recipients	2010	Census—SNAP
Some college	Percentage of people (age 25–44) with some post-secondary education	2007–2011	Census—ACS
Female-headed households	Percentage of female-headed households	2010	Census (Decennial)
Unemployment	Percentage of population (age 16+) unemployed but seeking work	2011	BLS
Violent crime	Number of reported violent crime offenses per 100,000 persons	2008–2010	FBI—UCR

Abbreviations: ACS: American Community Survey; BLS: Bureau of Labor Statistics; BRFSS: Behavioral Risk Factor Surveillance System; CBP: County Business Patterns; CDC: Centers for Disease Control and Prevention; DAHC: Dartmouth Atlas of Health Care; DIA: Diabetes Interactive Atlas; FEA: Food Environment Atlas; HRSA: Health Resources and Services Administration; NCHS: National Center for Health Statistics; PE: Population Estimates; SAHIE: Small Area Health Insurance Estimates; SAIPE: Small Area Income and Poverty Estimates; SEER: Surveillance, Epidemiology, and End Results; SNAP: Supplemental Nutrition Assistance Program; UCR: Uniform Crime Reporting; USDA: U.S. Department of Agriculture. Note: Area Deprivation Index was obtained using the ‘sociome’ R package; Community Health Centers, Poverty, Receipt of SNAP benefits, and Female-headed households were obtained from HRSA—Area Health Resources Files; All other variables were obtained from County Health Rankings and Roadmaps.

**Table 2 cancers-15-01006-t002:** Patient characteristics of EOCRC and LOCRC in the study area.

	EOCRC (Age < 50)	LOCRC (Age 50+)	*p*-Value
**N**	136,065	1,141,775	
**Incidence rate ***	6.9	134.7	
**Sex (%)**			<0.0001
Male	71,127 (52.3)	588,623 (51.6)	
Female	64,938 (47.7)	553,152 (48.4)	
**Race/Ethnicity (%)**			<0.0001
Non-Hispanic White	79,807 (58.7)	824,210 (72.2)	
Non-Hispanic Black	18,953 (13.9)	127,910 (11.2)	
Non-Hispanic American Indian/Alaska Native	711 (0.5)	3632 (0.3)	
Non-Hispanic Asian or Pacific Islander	10,252 (7.5)	62,769 (5.5)	
Hispanic (All Races)	25,240 (18.5)	118,234 (10.4)	
Non-Hispanic Unknown Race	1102 (0.8)	5020 (0.4)	
**Year of Diagnosis (%)**			<0.0001
2000–2004	29,908 (22.0)	306,441 (26.8)	
2005–2009	32,582 (23.9)	287,722 (25.2)	
2010–2014	34,203 (25.1)	271,980 (23.8)	
2015–2019	39,372 (28.9)	275,632 (24.1)	
**Tumor Site (%)**			<0.0001
Colon excluding Sigmoid Colon	55,063 (40.6)	606,845 (53.1)	
Sigmoid Colon	28,722 (21.1)	216,986 (19.0)	
Rectum and Rectosigmoid Junction	52,280 (38.4)	317,944 (27.9)	

* Age-adjusted incidence rate per 100,000 persons. Abbreviations. EOCRC: early-onset colorectal cancer; LOCRC: late-onset colorectal cancer; NOS: not otherwise specified.

## Data Availability

The data can be shared upon request.

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
