# Peer review of "Geographic Variation and Risk Factor Association of Early Versus Late Onset Colorectal Cancer"

_cancers, 2023, doi:10.3390/cancers15041006_

Round 1

Reviewer 1 Report

This study does a good job at comparing the risk factors between EOCRC and LOCRC. Using the SEER data for 2000-2019 provided an adequate population size. The variables included in the analysis were appropriate and meaningful. Division of the colonic anatomy and incidence rates was helpful.

The discontinuous US regions analyzed decreased the interpretations of geographic contributions to the incidence of CRC.

This manuscript was clearly written and relevant to the management of CRC. The method of analysis was appropriate and scientifically sound. The tables and figures were excellently presented and facilitated easy interpretation. Discussion of the data was well formatted and concise. Conclusions were supported by study data and analysis.

Author Response

We sincerely appreciate the reviewer for all the positive comments! 

Reviewer 2 Report

In this manuscript, the authors analyze the differences in the incidence of colon and rectal cancer from both a geographical point of view and age of onset, using data extracted from active SEER in 1085 counties in 16 US states.

In addition to the detailed data with the characteristics of the tumors observed provided by the SEER in each county, data on explanatory variables available from various statistical sources concerning other health conditions (obesity, diabetes), lifestyles, welfare services, environmental conditions were considered.

The authors aggregated data from counties in which fewer than 16 incident cases were recorded to protect patient confidentiality and reduce the instability of incidence data.

The authors used a machine learning method, random forest analysis, to identify the most relevant explanatory variables and then propose a ranking of them.

This method of analysis is the innovative and very interesting feature of the work for which I believe it is necessary for the authors to explain it in a very detailed way to allow readers to understand the procedure. Furthermore, it is necessary to explain the method in detail to underline the potential of these types of analyzes to recognize relationships between the data collected in the field of scientific research and to propose interpretative models.

The authors expose the variables which influence the early or late onset of colorectal cancer with the procedure and the data used. Given the automatism of the method, the authors comment on the results obtained with the forest analysis in the discussion and this aspect must be treated with great attention by those who use these methods in which it is necessary to evaluate all aspects of scientific reasoning, among others, those of reproducibility and replicability of the results, but in particular their plausibility.

Author Response

We sincerely appreciate the reviewer for all the positive comments! Please see the attached for our response to the reviewer's comments.

Reviewer 3 Report

This large population-based study investigated the risk factors for EOCRC and LOCRC, using machine learning software.

This approach is original.

This study highlights differences in risk factors for these 2 types of CRC and geographical disparities.

These results will make it possible to adapt prevention and screening measures in these populations.

The paper is very clear, the methodology well explained and the discussion explains well the limitations of this study.

Author Response

(The authors gave the same response as above.)
